# Poly-encoders: architectures and pre-training strategies for fast and accurate multi-sentence scoring

**Samuel Humeau**[*]**, Kurt Shuster**[*]**, Marie-Anne Lachaux, Jason Weston**
Facebook AI Research
{samuelhumeau,kshuster,malachaux,jase}@fb.com

## Abstract

The use of deep pre-trained transformers has led to remarkable progress in a number of applications (Devlin et al., 2019). For tasks that make pairwise comparisons between sequences, matching a given input with a corresponding label, two approaches are common: *Cross-encoders* performing full self-attention over the pair and *Bi-encoders* encoding the pair separately. The former often performs better, but is too slow for practical use. In this work, we develop a new transformer architecture, the *Poly-encoder*, that learns global rather than token level self-attention features. We perform a detailed comparison of all three approaches, including what pre-training and fine-tuning strategies work best. We show our models achieve state-of-the-art results on four tasks; that Poly-encoders are faster than Cross-encoders and more accurate than Bi-encoders; and that the best results are obtained by pre-training on large datasets similar to the downstream tasks.

## 1 Introduction

Recently, substantial improvements to state-of-the-art benchmarks on a variety of language understanding tasks have been achieved through the use of deep pre-trained language models followed by fine-tuning (Devlin et al., 2019). In this work we explore improvements to this approach for the class of tasks that require multi-sentence scoring: given an input context, score a set of candidate labels, a setup common in retrieval and dialogue tasks, amongst others. Performance in such tasks has to be measured via two axes: prediction quality and prediction speed, as scoring many candidates can be prohibitively slow.

The current state-of-the-art focuses on using BERT models for pre-training (Devlin et al., 2019), which employ large text corpora on general subjects: Wikipedia and the Toronto Books Corpus (Zhu et al., 2015). Two classes of fine-tuned architecture are typically built on top: Bi-encoders and Cross-encoders. Cross-encoders (Wolf et al., 2019; Vig & Ramea, 2019), which perform full (cross) self-attention over a given input and label candidate, tend to attain much higher accuracies than their counterparts, Bi-encoders (Mazaré et al., 2018; Dinan et al., 2019), which perform self-attention over the input and candidate label separately and combine them at the end for a final representation. As the representations are separate, Bi-encoders are able to cache the encoded candidates, and reuse these representations for each input resulting in fast prediction times. Cross-encoders must recompute the encoding for each input and label; as a result, they are prohibitively slow at test time.

In this work, we provide novel contributions that improve both the quality and speed axes over the current state-of-the-art. We introduce the Poly-encoder, an architecture with an additional learnt attention mechanism that represents more global features from which to perform self-attention, resulting in performance gains over Bi-encoders and large speed gains over Cross-Encoders. To pre-train our architectures, we show that choosing abundant data more similar to our downstream task also brings significant gains over BERT pre-training. This is true across all different architecture choices and downstream tasks we try.

We conduct experiments comparing the new approaches, in addition to analysis of what works best for various setups of existing methods, on four existing datasets in the domains of dialogue and information retrieval (IR), with pre-training strategies based on Reddit (Mazaré et al., 2018) compared

---

[*] Joint First Authors.

to Wikipedia/Toronto Books (i.e., BERT). We obtain a new state-of-the-art on all four datasets with our best architectures and pre-training strategies, as well as providing practical implementations for real-time use. Our code and models will be released open-source.

## 2 RELATED WORK

The task of scoring candidate labels given an input context is a classical problem in machine learning. While multi-class classification is a special case, the more general task involves candidates as structured objects rather than discrete classes; in this work we consider the inputs and the candidate labels to be sequences of text.

There is a broad class of models that map the input and a candidate label separately into a common feature space wherein typically a dot product, cosine or (parameterized) non-linearity is used to measure their similarity. We refer to these models as *Bi-encoders*. Such methods include vector space models (Salton et al., 1975), LSI (Deerwester et al., 1990), supervised embeddings (Bai et al., 2009; Wu et al., 2018) and classical siamese networks (Bromley et al., 1994). For the next utterance prediction tasks we consider in this work, several Bi-encoder neural approaches have been considered, in particular Memory Networks (Zhang et al., 2018a) and Transformer Memory networks (Dinan et al., 2019) as well as LSTMs (Lowe et al., 2015) and CNNs (Kadlec et al., 2015) which encode input and candidate label separately. A major advantage of Bi-encoder methods is their ability to cache the representations of a large, fixed candidate set. Since the candidate encodings are independent of the input, Bi-encoders are very efficient during evaluation.

Researchers have also studied a more rich class of models we refer to as *Cross-encoders*, which make no assumptions on the similarity scoring function between input and candidate label. Instead, the concatenation of the input and a candidate serve as a new input to a nonlinear function that scores their match based on any dependencies it wants. This has been explored with Sequential Matching Network CNN-based architectures (Wu et al., 2017), Deep Matching Networks (Yang et al., 2018), Gated Self-Attention (Zhang et al., 2018b), and most recently transformers (Wolf et al., 2019; Vig & Ramea, 2019; Urbanek et al., 2019). For the latter, concatenating the two sequences of text results in applying self-attention at every layer. This yields rich interactions between the input context and the candidate, as every word in the candidate label can attend to every word in the input context, and vice-versa. Urbanek et al. (2019) employed pre-trained BERT models, and fine-tuned both Bi- and Cross-encoders, explicitly comparing them on dialogue and action tasks, and finding that Cross-encoders perform better. However, the performance gains come at a steep computational cost. Cross-encoder representations are much slower to compute, rendering some applications infeasible.

## 3 TASKS

We consider the tasks of sentence selection in dialogue and article search in IR. The former is a task extensively studied and recently featured in two competitions: the Neurips ConvAI2 competition (Dinan et al., 2020), and the DSTC7 challenge, Track 1 (Yoshino et al., 2019; Jonathan K. Kummerfeld & Lasecki, 2018; Chulaka Gunasekara & Lasecki, 2019). We compare on those two tasks and in addition, we also test on the popular Ubuntu V2 corpus (Lowe et al., 2015). For IR, we use the Wikipedia Article Search task of Wu et al. (2018).

The ConvAI2 task is based on the Persona-Chat dataset (Zhang et al., 2018a) which involves dialogues between pairs of speakers. Each speaker is given a persona, which is a few sentences that describe a character they will imitate, e.g. "I love romantic movies", and is instructed to get to know the other. Models should then condition their chosen response on the dialogue history and the lines of persona. As an automatic metric in the competition, for each response, the model has to pick the correct annotated utterance from a set of 20 choices, where the remaining 19 were other randomly chosen utterances from the evaluation set. Note that in a final system however, one would retrieve from the entire training set of over 100k utterances, but this is avoided for speed reasons in common evaluation setups. The best performing competitor out of 23 entrants in this task achieved 80.7% accuracy on the test set utilizing a pre-trained Transformer fine-tuned for this task (Wolf et al., 2019).

The DSTC7 challenge (Track 1) consists of conversations extracted from Ubuntu chat logs, where one partner receives technical support for various Ubuntu-related problems from the other. The

best performing competitor (with 20 entrants in Track 1) in this task achieved 64.5% R@1 (Chen & Wang, 2019). Ubuntu V2 is a similar but larger popular corpus, created before the competition (Lowe et al., 2015); we report results for this dataset as well, as there are many existing results on it.

Finally, we evaluate on Wikipedia Article Search (Wu et al., 2018). Using the 2016-12-21 dump of English Wikipedia (~5M articles), the task is given a sentence from an article as a search query, find the article it came from. Evaluation ranks the true article (minus the sentence) against 10,000 other articles using retrieval metrics. This mimics a web search like scenario where one would like to search for the most relevant articles (web documents). The best reported method is the learning-to-rank embedding model, StarSpace, which outperforms fastText, SVMs, and other baselines.

We summarize all four datasets and their statistics in Table 1.

|  | ConvAI2 | DTSC7 | Ubuntu V2 | Wiki Article Search |
|---|---|---|---|---|
| Train Ex. | 131,438 | 100,000 | 1,000,000 | 5,035,182 |
| Valid Ex. | 7,801 | 10,000 | 19,560 | 9,921 |
| Test Ex. | 6634 | 5,000 | 18,920 | 9,925 |
| Eval Cands per Ex. | 20 | 100 | 10 | 10,001 |

Table 1: Datasets used in this paper.

## 4 METHODS

In this section we describe the various models and methods that we explored.

### 4.1 TRANSFORMERS AND PRE-TRAINING STRATEGIES

**Transformers** Our Bi-, Cross-, and Poly-encoders, described in sections 4.2, 4.3 and 4.4 respectively, are based on large pre-trained transformer models with the same architecture and dimension as BERT-base (Devlin et al., 2019), which has 12 layers, 12 attention heads, and a hidden size of 768. As well as considering the BERT pre-trained weights, we also explore our own pre-training schemes. Specifically, we pre-train two more transformers from scratch using the exact same architecture as BERT-base. One uses a similar training setup as in BERT-base, training on 150 million of examples of [INPUT, LABEL] extracted from Wikipedia and the Toronto Books Corpus, while the other is trained on 174 million examples of [INPUT, LABEL] extracted from the online platform Reddit (Mazaré et al., 2018), which is a dataset more adapted to dialogue. The former is performed to verify that reproducing a BERT-like setting gives us the same results as reported previously, while the latter tests whether pre-training on data more similar to the downstream tasks of interest helps. For training both new setups we used XLM (Lample & Conneau, 2019).

**Input Representation** Our pre-training input is the concatenation of input and label [INPUT,LABEL], where both are surrounded with the special token [S], following Lample & Conneau (2019). When pre-training on Reddit, the input is the context, and the label is the next utterance. When pre-training on Wikipedia and Toronto Books, as in Devlin et al. (2019), the input is one sentence and the label the next sentence in the text. Each input token is represented as the sum of three embeddings: the token embedding, the position (in the sequence) embedding and the segment embedding. Segments for input tokens are 0, and for label tokens are 1.

**Pre-training Procedure** Our pre-training strategy involves training with a masked language model (MLM) task identical to the one in Devlin et al. (2019). In the pre-training on Wikipedia and Toronto Books we add a next-sentence prediction task identical to BERT training. In the pre-training on Reddit, we add a next-utterance prediction task, which is slightly different from the previous one as an utterance can be composed of several sentences. During training 50% of the time the candidate is the actual next sentence/utterance and 50% of the time it is a sentence/utterance randomly taken from the dataset. We alternate between batches of the MLM task and the next-sentence/next-utterance prediction task. Like in Lample & Conneau (2019) we use the Adam optimizer with learning rate of 2e-4, $\beta_1 = 0.9$, $\beta_2 = 0.98$, no L2 weight decay, linear learning rate warmup, and inverse square root decay of the learning rate. We use a dropout probability of 0.1 on all layers, and

a batch of 32000 tokens composed of concatenations [INPUT, LABEL] with similar lengths. We train the model on 32 GPUs for 14 days.

**Fine-tuning**  After pre-training, one can then fine-tune for the multi-sentence selection task of choice, in our case one of the four tasks from Section 3. We consider three architectures with which we fine-tune the transformer: the Bi-encoder, Cross-encoder and newly proposed Poly-encoder.

## 4.2  BI-ENCODER

In a Bi-encoder, both the input context and the candidate label are encoded into vectors:

$$y_{ctxt} = red(T_1(ctxt)) \qquad y_{cand} = red(T_2(cand))$$

where $T_1$ and $T_2$ are two transformers that have been pre-trained following the procedure described in 4.1; they initially start with the same weights, but are allowed to update separately during fine-tuning. $T(x) = h_1, .., h_N$ is the output of a transformer T and $red(\cdot)$ is a function that reduces that sequence of vectors into one vector. As the input and the label are encoded separately, segment tokens are 0 for both. To resemble what is done during our pre-training, both the input and label are surrounded by the special token [S] and therefore $h_1$ corresponds to [S].

We considered three ways of reducing the output into one representation via $red(\cdot)$: choose the first output of the transformer (corresponding to the special token [S]), compute the average over all outputs or the average over the first $m \leq N$ outputs. We compare them in Table 7 in the Appendix. We use the first output of the transformer in our experiments as it gives slightly better results.

**Scoring**  The score of a candidate $cand_i$ is given by the dot-product $s(ctxt, cand_i) = y_{ctxt} \cdot y_{cand_i}$. The network is trained to minimize a cross-entropy loss in which the logits are $y_{ctxt} \cdot y_{cand_1}, ..., y_{ctxt} \cdot y_{cand_n}$, where $cand_1$ is the correct label and the others are chosen from the training set. Similar to Mazaré et al. (2018), during training we consider the other labels in the batch as negatives. This allows for much faster training, as we can reuse the embeddings computed for each candidate, and also use a larger batch size; e.g., in our experiments on ConvAI2, we were able to use batches of 512 elements.

**Inference speed**  In the setting of retrieval over known candidates, a Bi-encoder allows for the precomputation of the embeddings of all possible candidates of the system. After the context embedding $y_{ctxt}$ is computed, the only operation remaining is a dot product between $y_{ctxt}$ and every candidate embedding, which can scale to millions of candidates on a modern GPU, and potentially billions using nearest-neighbor libraries such as FAISS (Johnson et al., 2019).

## 4.3  CROSS-ENCODER

The Cross-encoder allows for rich interactions between the input context and candidate label, as they are jointly encoded to obtain a final representation. Similar to the procedure in pre-training, the context and candidate are surrounded by the special token [S] and concatenated into a single vector, which is encoded using one transformer. We consider the first output of the transformer as the context-candidate embedding:

$$y_{ctxt,cand} = h_1 = first(T(ctxt, cand))$$

where $first$ is the function that takes the first vector of the sequence of vectors produced by the transformer. By using a single transformer, the Cross-encoder is able to perform self-attention between the context and candidate, resulting in a richer extraction mechanism than the Bi-encoder. As the candidate label can attend to the input context during the layers of the transformer, the Cross-encoder can produce a candidate-sensitive input representation, which the Bi-encoder cannot. For example, this allows it to select useful input features per candidate.

**Scoring**  To score one candidate, a linear layer $W$ is applied to the embedding $y_{ctxt,cand}$ to reduce it from a vector to a scalar:

$$s(ctxt, cand_i) = y_{ctxt,cand_i} W$$

Similarly to what is done for the Bi-encoder, the network is trained to minimize a cross entropy loss where the logits are $s(ctxt, cand_1), ..., s(ctxt, cand_n)$, where $cand_1$ is the correct candidate and the

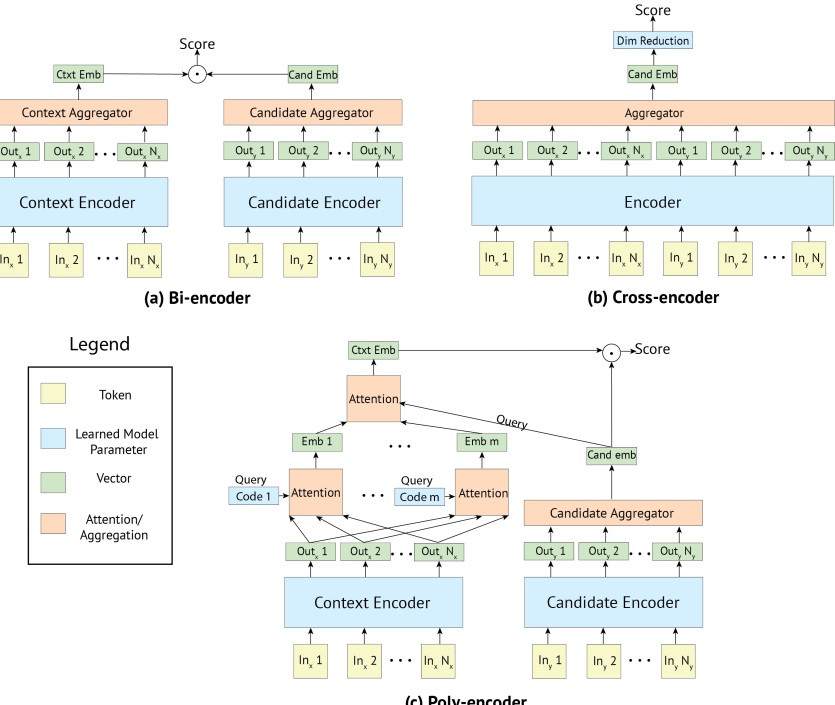

Figure 1: Diagrams of the three model architectures we consider. (a) The Bi-encoder encodes the context and candidate separately, allowing for the caching of candidate representations during inference. (b) The Cross-encoder jointly encodes the context and candidate in a single transformer, yielding richer interactions between context and candidate at the cost of slower computation. (c) The Poly-encoder combines the strengths of the Bi-encoder and Cross-encoder by both allowing for caching of candidate representations and adding a final attention mechanism between global features of the input and a given candidate to give richer interactions before computing a final score.

others are negatives taken from the training set. Unlike in the Bi-encoder, we cannot recycle the other labels of the batch as negatives, so we use external negatives provided in the training set. The Cross-encoder uses much more memory than the Bi-encoder, resulting in a much smaller batch size.

**Inference speed**   Unfortunately, the Cross-encoder does not allow for precomputation of the candidate embeddings. At inference time, every candidate must be concatenated with the input context and must go through a forward pass of the entire model. Thus, this method cannot scale to a large amount of candidates. We discuss this bottleneck further in Section 5.4.

### 4.4    POLY-ENCODER

The Poly-encoder architecture aims to get the best of both worlds from the Bi- and Cross-encoder. A given candidate label is represented by one vector as in the Bi-encoder, which allows for caching candidates for fast inference time, while the input context is jointly encoded with the candidate, as in the Cross-encoder, allowing the extraction of more information.

The Poly-encoder uses two separate transformers for the context and label like a Bi-encoder, and the candidate is encoded into a single vector $y_{cand_i}$. As such, the Poly-encoder method can be implemented using a precomputed cache of encoded responses. However, the input context, which is typically much longer than a candidate, is represented with $m$ vectors $(y_{ctxt}^1..y_{ctxt}^m)$ instead of just one as in the Bi-encoder, where $m$ will influence the inference speed. To obtain these $m$ global features that represent the input, we learn $m$ context codes $(c_1, ..., c_m)$, where $c_i$ extracts representation $y_{ctxt}^i$ by attending over all the outputs of the previous layer. That is, we obtain $y_{ctxt}^i$ using:

$$y_{ctxt}^i = \sum_j w_j^{c_i} h_j \qquad \text{where} \qquad (w_1^{c_i}, .., w_N^{c_i}) = \text{softmax}(c_i \cdot h_1, .., c_i \cdot h_N)$$

The *m* context codes are randomly initialized, and learnt during finetuning.

Finally, given our *m* global context features, we attend over them using $y_{cand_i}$ as the query:

$$y_{ctxt} = \sum_i w_i y_{ctxt}^i \qquad \text{where} \qquad (w_1, .., w_m) = \text{softmax}(y_{cand_i} \cdot y_{ctxt}^1, .., y_{cand_i} \cdot y_{ctxt}^m)$$

The final score for that candidate label is then $y_{ctxt} \cdot y_{cand_i}$ as in a Bi-encoder. As $m < N$, where $N$ is the number of tokens, and the context-candidate attention is only performed at the top layer, this is far faster than the Cross-encoder's full self-attention.

## 5 EXPERIMENTS

We perform a variety of experiments to test our model architectures and training strategies over four tasks. For metrics, we measure Recall@$k$ where each test example has $C$ possible candidates to select from, abbreviated to R@$k/C$, as well as mean reciprocal rank (MRR).

### 5.1 BI-ENCODERS AND CROSS-ENCODERS

We first investigate fine-tuning the Bi- and Cross-encoder architectures initialized with the weights provided by Devlin et al. (2019), studying the choice of other hyperparameters (we explore our own pre-training schemes in section 5.3). In the case of the Bi-encoder, we can use a large number of negatives by considering the other batch elements as negative training samples, avoiding recomputation of their embeddings. On 8 Nvidia Volta v100 GPUs and using half-precision operations (i.e. float16 operations), we can reach batches of 512 elements on ConvAI2. Table 2 shows that in this setting, we obtain higher performance with a larger batch size, i.e. more negatives, where 511 negatives yields the best results. For the other tasks, we keep the batch size at 256, as the longer sequences in those datasets uses more memory. The Cross-encoder is more computationally intensive, as the embeddings for the (context, candidate) pair must be recomputed each time. We thus limit its batch size to 16 and provide negatives random samples from the training set. For DSTC7 and Ubuntu V2, we choose 15 such negatives; For ConvAI2, the dataset provides 19 negatives.

| Negatives | 31 | 63 | 127 | 255 | 511 |
|---|---|---|---|---|---|
| R@1/20 | 81.0 | 81.7 | 82.3 | 83.0 | **83.3** |

Table 2: Validation performance on ConvAI2 after fine-tuning a Bi-encoder pre-trained with BERT, averaged over 5 runs. The batch size is the number of training negatives + 1 as we use the other elements of the batch as negatives during training.

The above results are reported with Bi-encoder aggregation based on the first output. Choosing the average over all outputs instead is very similar but slightly worse (83.1, averaged over 5 runs). We also tried to add further non-linearities instead of the inner product of the two representations, but could not obtain improved results over the simpler architecture (results not shown).

We tried two optimizers: Adam (Kingma & Ba, 2015) with weight decay of 0.01 (as recommended by (Devlin et al., 2019)) and Adamax (Kingma & Ba, 2015) without weight decay; based on validation set performance, we choose to fine-tune with Adam when using the BERT weights. The learning rate is initialized to 5e-5 with a warmup of 100 iterations for Bi- and Poly-encoders, and 1000 iterations for the Cross-encoder. The learning rate decays by a factor of 0.4 upon plateau of the loss evaluated on the valid set every half epoch. In Table 3 we show validation performance when fine-tuning various layers of the weights provided by (Devlin et al., 2019), using Adam with decay optimizer. Fine-tuning the entire network is important, with the exception of the word embeddings.

With the setups described above, we fine-tune the Bi- and Cross-encoders on the datasets, and report the results in Table 4. On the first three tasks, our Bi-encoders and Cross-encoders outperform the best existing approaches in the literature when we fine-tune from BERT weights. E.g., the Bi-encoder reaches 81.7% R@1 on ConvAI2 and 66.8% R@1 on DSTC7, while the Cross-encoder achieves higher scores of 84.8% R@1 on ConvAI2 and 67.4% R@1 on DSTC7. Overall, Cross-encoders outperform all previous approaches on the three dialogue tasks, including our Bi-encoders (as expected). We do not report fine-tuning of BERT for Wikipedia IR as we cannot guarantee the

| Fine-tuned parameters | Bi-encoder | Cross-encoder |
|---|---|---|
| Top layer | 74.2 | 80.6 |
| Top 4 layers | 82.0 | 86.3 |
| All but Embeddings | **83.3** | **87.3** |
| Every Layer | 83.0 | 86.6 |

Table 3: Validation performance (R@1/20) on ConvAI2 using pre-trained weights of BERT-base with different parameters fine-tuned. Average over 5 runs (Bi-encoders) or 3 runs (Cross-encoders).

test set is not part of the pre-training for that dataset. In addition, Cross-encoders are also too slow to evaluate on the evaluation setup of that task, which has 10k candidates.

| Dataset | ConvAI2 | DSTC 7 | | Ubuntu v2 | | Wikipedia IR |
|---|---|---|---|---|---|---|
| split | test | test | | test | | test |
| metric | R@1/20 | R@1/100 | MRR | R@1/10 | MRR | R@1/10001 |
| (Wolf et al., 2019) | 80.7 | | | | | |
| (Gu et al., 2018) | - | 60.8 | 69.1 | - | - | - |
| (Chen & Wang, 2019) | - | 64.5 | 73.5 | - | - | - |
| (Yoon et al., 2018) | - | - | - | 65.2 | - | - |
| (Dong & Huang, 2018) | - | - | - | 75.9 | 84.8 | - |
| (Wu et al., 2018) | - | - | - | - | - | 56.8 |
| pre-trained BERT weights from (Devlin et al., 2019) - Toronto Books + Wikipedia | | | | | | |
| Bi-encoder | 81.7 ± 0.2 | 66.8 ± 0.7 | 74.6 ± 0.5 | 80.6 ± 0.4 | 88.0 ± 0.3 | - |
| Poly-encoder 16 | 83.2 ± 0.1 | 67.8 ± 0.3 | 75.1 ± 0.2 | 81.2 ± 0.2 | 88.3 ± 0.1 | - |
| Poly-encoder 64 | 83.7 ± 0.2 | 67.0 ± 0.9 | 74.7 ± 0.6 | 81.3 ± 0.2 | 88.4 ± 0.1 | - |
| Poly-encoder 360 | 83.7 ± 0.2 | 68.9 ± 0.4 | 76.2 ± 0.2 | 80.9 ± 0.0 | 88.1 ± 0.1 | - |
| Cross-encoder | 84.8 ± 0.3 | 67.4 ± 0.7 | 75.6 ± 0.4 | 82.8 ± 0.3 | 89.4 ± 0.2 | - |
| Our pre-training on Toronto Books + Wikipedia | | | | | | |
| Bi-encoder | 82.0 ± 0.1 | 64.5 ± 0.5 | 72.6 ± 0.4 | 80.8 ± 0.5 | 88.2 ± 0.4 | - |
| Poly-encoder 16 | 82.7 ± 0.1 | 65.3 ± 0.9 | 73.2 ± 0.7 | 83.4 ± 0.2 | 89.9 ± 0.1 | - |
| Poly-encoder 64 | 83.3 ± 0.1 | 65.8 ± 0.7 | 73.5 ± 0.5 | 83.4 ± 0.1 | 89.9 ± 0.0 | - |
| Poly-encoder 360 | 83.8 ± 0.1 | 65.8 ± 0.7 | 73.6 ± 0.6 | 83.7 ± 0.0 | 90.1 ± 0.0 | - |
| Cross-encoder | 84.9 ± 0.3 | 65.3 ± 1.0 | 73.8 ± 0.6 | 83.1 ± 0.7 | 89.7 ± 0.5 | - |
| Our pre-training on Reddit | | | | | | |
| Bi-encoder | 84.8 ± 0.1 | 70.9 ± 0.5 | 78.1 ± 0.3 | 83.6 ± 0.7 | 90.1 ± 0.4 | 71.0 |
| Poly-encoder 16 | 86.3 ± 0.3 | 71.6 ± 0.6 | 78.4 ± 0.4 | 86.0 ± 0.1 | 91.5 ± 0.1 | 71.5 |
| Poly-encoder 64 | 86.5 ± 0.2 | 71.2 ± 0.8 | 78.2 ± 0.7 | 85.9 ± 0.1 | 91.5 ± 0.1 | 71.3 |
| Poly-encoder 360 | 86.8 ± 0.1 | 71.4 ± 1.0 | 78.3 ± 0.7 | 85.9 ± 0.1 | 91.5 ± 0.0 | **71.8** |
| Cross-encoder | **87.9 ± 0.2** | **71.7 ± 0.3** | **79.0 ± 0.2** | **86.5 ± 0.1** | **91.9 ± 0.0** | - |

Table 4: Test performance of Bi-, Poly- and Cross-encoders on our selected tasks.

## 5.2 POLY-ENCODERS

We train the Poly-encoder using the same batch sizes and optimizer choices as in the Bi-encoder experiments. Results are reported in Table 4 for various values of $m$ context vectors.

The Poly-encoder outperforms the Bi-encoder on all the tasks, with more codes generally yielding larger improvements. Our recommendation is thus to use as large a code size as compute time allows (see Sec. 5.4). On DSTC7, the Poly-encoder architecture with BERT pretraining reaches 68.9% R1 with 360 intermediate context codes; this actually outperforms the Cross-encoder result (67.4%) and is noticeably better than our Bi-encoder result (66.8%). Similar conclusions are found on Ubuntu V2 and ConvAI2, although in the latter Cross-encoders give slightly better results.

We note that since reporting our results, the authors of Li et al. (2019) have conducted a human evaluation study on ConvAI2, in which our Poly-encoder architecture outperformed all other models compared against, both generative and retrieval based, including the winners of the competition.

| | Scoring time (ms) | | | |
|---|---|---|---|---|
| | CPU | | GPU | |
| Candidates | 1k | 100k | 1k | 100k |
| Bi-encoder | 115 | 160 | 19 | 22 |
| Poly-encoder 16 | 122 | 678 | 18 | 38 |
| Poly-encoder 64 | 126 | 692 | 23 | 46 |
| Poly-encoder 360 | 160 | 837 | 57 | 88 |
| Cross-encoder | 21.7k | 2.2M* | 2.6k | 266k* |

Table 5: Average time in milliseconds to predict the next dialogue utterance from $C$ possible candidates on ConvAI2. * are inferred.

## 5.3 DOMAIN-SPECIFIC PRE-TRAINING

We fine-tune our Reddit-pre-trained transformer on all four tasks; we additionally fine-tune a transformer that was pre-trained on the same datasets as BERT, specifically Toronto Books + Wikipedia. When using our pre-trained weights, we use the Adamax optimizer and optimize all the layers of the transformer including the embeddings. As we do not use weight decay, the weights of the final layer are much larger than those in the final layer of BERT; to avoid saturation of the attention layer in the Poly-encoder, we re-scaled the last linear layer so that the standard deviation of its output matched that of BERT, which we found necessary to achieve good results. We report results of fine-tuning with our pre-trained weights in Table 4. We show that pre-training on Reddit gives further state-of-the-art performance over our previous results with BERT, a finding that we see for all three dialogue tasks, and all three architectures.

The results obtained with fine-tuning on our own transformers pre-trained on Toronto Books + Wikipedia are very similar to those obtained with the original BERT weights, indicating that the choice of dataset used to pre-train the models impacts the final results, not some other detail in our training. Indeed, as the two settings pre-train with datasets of similar size, we can conclude that choosing a pre-training task (e.g. dialogue data) that is similar to the downstream tasks of interest (e.g. dialogue) is a likely explanation for these performance gains, in line with previous results showing multi-tasking with similar tasks is more useful than with dissimilar ones (Caruana, 1997).

## 5.4 INFERENCE SPEED

An important motivation for the Poly-encoder architecture is to achieve better results than the Bi-encoder while also performing at a reasonable speed. Though the Cross-encoder generally yields strong results, it is prohibitively slow. We perform speed experiments to determine the trade-off of improved performance from the Poly-encoder. Specifically, we predict the next utterance for 100 dialogue examples in the ConvAI2 validation set, where the model scores $C$ candidates (in this case, chosen from the training set). We perform these experiments on both CPU-only and GPU setups. CPU computations were run on an 80 core Intel Xeon processor CPU E5-2698. GPU computations were run on a single Nvidia Quadro GP100 using cuda 10.0 and cudnn 7.4.

We show the average time per example for each architecture in Table 5. The difference in timing between the Bi-encoder and the Poly-encoder architectures is rather minimal when there are only 1000 candidates for the model to consider. The difference is more pronounced when considering 100k candidates, a more realistic setup, as we see a 5-6x slowdown for the Poly-encoder variants. Nevertheless, both models are still tractable. The Cross-encoder, however, is 2 orders of magnitude slower than the Bi-encoder and Poly-encoder, rendering it intractable for real-time inference, e.g. when interacting with a dialogue agent, or retrieving from a large set of documents. Thus, Poly-encoders, given their desirable performance and speed trade-off, are the preferred method.

We additionally report training times in the Appendix, Table 6. Poly-encoders also have the benefit of being 3-4x faster to train than Cross-encoders (and are similar in training time to Bi-encoders).

## 6 CONCLUSION

In this paper we present new architectures and pre-training strategies for deep bidirectional transformers in candidate selection tasks. We introduced the Poly-encoder method, which provides a mechanism for attending over the context using the label candidate, while maintaining the ability to precompute each candidate's representation, which allows for fast real-time inference in a production setup, giving an improved trade off between accuracy and speed. We provided an experimental analysis of those trade-offs for Bi-, Poly- and Cross-encoders, showing that Poly-encoders are more accurate than Bi-encoders, while being far faster than Cross-encoders, which are impractical for real-time use. In terms of training these architectures, we showed that pre-training strategies more closely related to the downstream task bring strong improvements. In particular, pre-training from scratch on Reddit allows us to outperform the results we obtain with BERT, a result that holds for all three model architectures and all three dialogue datasets we tried. However, the methods introduced in this work are not specific to dialogue, and can be used for any task where one is scoring a set of candidates, which we showed for an information retrieval task as well.

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

## A    TRAINING TIME

We report the training time on 8 GPU Volta 100 for the 3 datasets considered and for 4 types of models in Table 6.

| Dataset | ConvAI2 | DSTC7 | UbuntuV2 |
|---|---|---|---|
| Bi-encoder | 2.0 | 4.9 | 7.9 |
| Poly-encoder 16 | 2.7 | 5.5 | 8.0 |
| Poly-encoder 64 | 2.8 | 5.7 | 8.0 |
| Cross-encoder64 | 9.4 | 13.5 | 39.9 |

Table 6: Training time in hours.

## B    REDUCTION LAYER IN BI-ENCODER

We provide in Table 7 the results obtained for different types of reductions on top of the Bi-encoder. Specifically we compare the Recall@1/20 on the ConvAI2 validation set when taking the first output of BERT, the average of the first 16 outputs, the average of the first 64 outputs and all of them except the first one ([S]).

| Setup | ConvAI2 valid Recall@1/20 |
|---|---|
| First output | 83.3 |
| Avg first 16 outputs | 82.9 |
| Avg first 64 outputs | 82.7 |
| Avg all outputs | 83.1 |

Table 7: Bi-encoder results on the ConvAI2 valid set for different choices of function $red(\cdot)$.

## C    ALTERNATIVE CHOICES FOR CONTEXT VECTORS

We considered a few other ways to derive the context vectors $(y^1_{ctxt}, ..., y^m_{ctxt})$ of the Poly-encoder from the output $(h^1_{ctxt}, ..., h^N_{ctxt})$ of the underlying transformer:

- Learn $m$ codes $(c_1, ..., c_m)$, where $c_i$ extracts representation $y^i_{ctxt}$ by attending over all the outputs $(h^1_{ctxt}, ..., h^N_{ctxt})$. This method is denoted "Poly-encoder (Learnt-codes)" or "Poly-encoder (Learnt-m)", and is the method described in section 4.4

- Consider the first $m$ outputs $(h^1_{ctxt}, ..., h^m_{ctxt})$. This method is denoted "Poly-encoder (First $m$ outputs)" or "Poly-encoder (First-m)". Note that when $N < m$, only $m$ vectors are considered.

- Consider the last $m$ outputs.

- Consider the last $m$ outputs concatenated with the first one, $h^1_{ctxt}$ which plays a particular role in BERT as it corresponds to the special token [S].

The performance of those four methods is evaluated on the validation set of Convai2 and DSTC7 and reported on Table 8. The first two methods are shown in Figure 2. We additionally provide the inference time for a given number of candidates coming from the Convai2 dataset on Table 9.

| Dataset | ConvAI2 | | DSTC 7 | |
|---|---|---|---|---|
| split | dev | test | dev | test |
| metric | R@1/20 | R@1/20 | R@1/100 | R@1/100 |
| (Wolf et al., 2019) | 82.1 | 80.7 | - | - |
| (Chen & Wang, 2019) | - | - | 57.3 | 64.5 |
| **1 Attention Code** | | | | |
| Learnt-codes | 81.9 ± 0.3 | 81.0 ± 0.1 | 56.2 ± 0.1 | 66.9 ± 0.7 |
| First $m$ outputs | 83.2 ± 0.2 | 81.5 ± 0.1 | 56.4 ± 0.3 | 66.8 ± 0.7 |
| Last $m$ outputs | 82.9 ± 0.1 | 81.0 ± 0.1 | 56.1 ± 0.4 | 67.2 ± 1.1 |
| Last $m$ outputs and $h_{ctxt}^1$ | - | - | - | - |
| **4 Attention Codes** | | | | |
| Learnt-codes | **83.8 ± 0.2** | **82.2 ± 0.5** | 56.5 ± 0.5 | 66.8 ± 0.7 |
| First $m$ outputs | 83.4 ± 0.2 | 81.6 ± 0.1 | **56.9 ± 0.5** | **67.2 ± 1.3** |
| Last $m$ outputs | 82.8 ± 0.2 | 81.3 ± 0.4 | 56.0 ± 0.5 | 65.8 ± 0.5 |
| Last $m$ outputs and $h_{ctxt}^1$ | 82.9 ± 0.1 | 81.4 ± 0.2 | 55.8 ± 0.3 | 66.1 ± 0.8 |
| **16 Attention Codes** | | | | |
| Learnt-codes | 84.4 ± 0.1 | 83.2 ± 0.1 | **57.7 ± 0.2** | **67.8 ± 0.3** |
| First $m$ outputs | **85.2 ± 0.1** | **83.9 ± 0.2** | 56.1 ± 1.7 | 66.8 ± 1.1 |
| Last $m$ outputs | 83.9 ± 0.2 | 82.0 ± 0.4 | 56.1 ± 0.3 | 66.2 ± 0.7 |
| Last $m$ outputs and $h_{ctxt}^1$ | 83.8 ± 0.3 | 81.7 ± 0.3 | 56.1 ± 0.3 | 66.6 ± 0.2 |
| **64 Attention Codes** | | | | |
| Learnt-codes | 84.9 ± 0.1 | 83.7 ± 0.2 | **58.3 ± 0.4** | 67.0 ± 0.9 |
| First $m$ outputs | **86.0 ± 0.2** | **84.2 ± 0.2** | 57.7 ± 0.6 | **67.1 ± 0.1** |
| Last $m$ outputs | 84.9 ± 0.3 | 82.9 ± 0.2 | 57.0 ± 0.2 | 66.5 ± 0.5 |
| Last $m$ outputs and $h_{ctxt}^1$ | 85.0 ± 0.2 | 83.2 ± 0.2 | 57.3 ± 0.3 | **67.1 ± 0.5** |
| **360 Attention Codes** | | | | |
| Learnt-codes | 85.3 ± 0.3 | 83.7 ± 0.2 | 57.7 ± 0.3 | **68.9 ± 0.4** |
| First $m$ outputs | **86.3 ± 0.1** | 84.6 ± 0.3 | 58.1 ± 0.4 | 66.8 ± 0.7 |
| Last $m$ outputs | **86.3 ± 0.1** | **84.7 ± 0.3** | 58.0 ± 0.4 | 68.1 ± 0.5 |
| Last $m$ outputs and $h_{ctxt}^1$ | 86.2 ± 0.3 | 84.5 ± 0.4 | **58.3 ± 0.4** | 68.0 ± 0.8 |

Table 8: Validation and test performance of Poly-encoder variants, with weights initialized from (Devlin et al., 2019). Scores are shown for ConvAI2 and DSTC 7 Track 1. Bold numbers indicate the highest performing variant within that number of codes.

| | Scoring time (ms) | | | |
|---|---|---|---|---|
| | CPU | | GPU | |
| Candidates | 1k | 100k | 1k | 100k |
| Bi-encoder | 115 | 160 | 19 | 22 |
| Poly-encoder (First $m$ outputs) 16 | 119 | 551 | 17 | 37 |
| Poly-encoder (First $m$ outputs) 64 | 124 | 570 | 17 | 39 |
| Poly-encoder (First $m$ outputs) 360 | 120 | 619 | 17 | 45 |
| Poly-encoder (Learnt-codes) 16 | 122 | 678 | 18 | 38 |
| Poly-encoder (Learnt-codes) 64 | 126 | 692 | 23 | 46 |
| Poly-encoder (Learnt-codes) 360 | 160 | 837 | 57 | 88 |
| Cross-encoder | 21.7k | 2.2M* | 2.6k | 266k* |

Table 9: Average time in milliseconds to predict the next dialogue utterance from $N$ possible candidates. * are inferred.

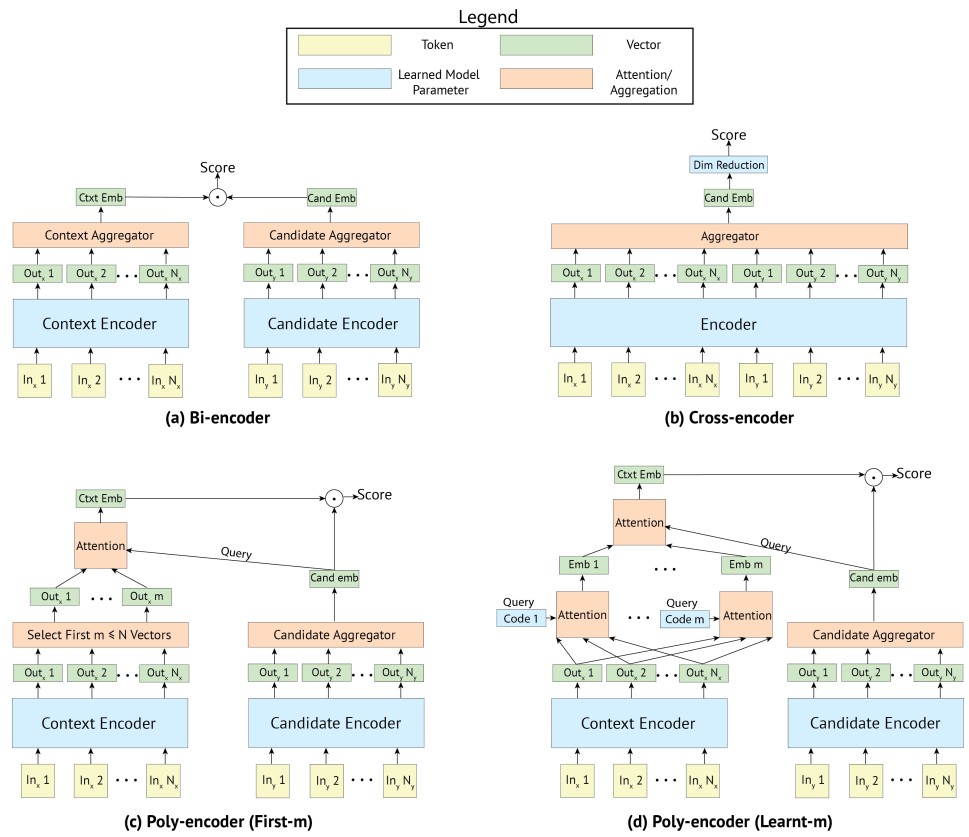

Figure 2: (a) The Bi-encoder (b) The Cross-encoder (c) The Poly-encoder with first $m$ vectors. (d) The Poly-encoder with $m$ learnt codes.

| Dataset | ConvAI2 | | DSTC 7 | | | | Ubuntu v2 | | | |
|---|---|---|---|---|---|---|---|---|---|---|
| split | dev | test | dev | test | | | dev | test | | |
| metric | R@1/20 | R@1/20 | R@1/100 | R@1/100 | R@10/100 | MRR | R@1/10 | R@1/10 | R@5/10 | MRR |
| Hugging Face (Wolf et al., 2019) | 82.1 | 80.7 | - | - | - | - | - | - | - | - |
| (Chen & Wang, 2019) | - | - | 57.3 | 64.5 | 90.2 | 73.5 | - | - | - | - |
| (Dong & Huang, 2018) | - | - | - | - | - | - | - | 75.9 | 97.3 | 84.8 |
| pre-trained weights from (Devlin et al., 2019) - Toronto Books + Wikipedia | | | | | | | | | | |
| Bi-encoder | 83.3 ± 0.2 | 81.7 ± 0.2 | 56.5 ± 0.4 | 66.8 ± 0.7 | 89.0 ± 1.0 | 74.6 ± 0.5 | 80.9 ± 0.6 | 80.6 ± 0.4 | 98.2 ± 0.1 | 88.0 ± 0.3 |
| Poly-encoder (First-m) 16 | 85.2 ± 0.1 | 83.9 ± 0.2 | 56.7 ± 0.2 | 67.0 ± 0.9 | 88.8 ± 0.3 | 74.6 ± 0.6 | 81.7 ± 0.5 | 81.4 ± 0.6 | 98.2 ± 0.1 | 88.5 ± 0.4 |
| Poly-encoder (Learnt-m) 16 | 84.4 ± 0.1 | 83.2 ± 0.1 | 57.7 ± 0.2 | 67.8 ± 0.3 | 88.6 ± 0.2 | 75.1 ± 0.2 | 81.5 ± 0.1 | 81.2 ± 0.2 | 98.2 ± 0.0 | 88.3 ± 0.1 |
| Poly-encoder (First-m) 64 | 86.0 ± 0.2 | 84.2 ± 0.2 | 57.1 ± 0.2 | 66.9 ± 0.7 | 89.1 ± 0.2 | 74.7 ± 0.4 | 82.2 ± 0.6 | 81.9 ± 0.5 | 98.4 ± 0.0 | 88.8 ± 0.3 |
| Poly-encoder (Learnt-m) 64 | 84.9 ± 0.1 | 83.7 ± 0.2 | 58.3 ± 0.4 | 67.0 ± 0.9 | 89.2 ± 0.2 | 74.7 ± 0.6 | 81.8 ± 0.1 | 81.3 ± 0.2 | 98.2 ± 0.1 | 88.4 ± 0.1 |
| Poly-encoder (First-m) 360 | 86.3 ± 0.1 | 84.6 ± 0.3 | 57.8 ± 0.5 | 67.0 ± 0.5 | 89.6 ± 0.9 | 75.0 ± 0.6 | 82.7 ± 0.4 | 82.2 ± 0.6 | 98.4 ± 0.1 | 89.0 ± 0.4 |
| Poly-encoder (Learnt-m) 360 | 85.3 ± 0.3 | 83.7 ± 0.2 | 57.7 ± 0.3 | 68.9 ± 0.4 | 89.9 ± 0.5 | 76.2 ± 0.2 | 81.5 ± 0.1 | 80.9 ± 0.1 | 98.1 ± 0.0 | 88.1 ± 0.1 |
| Cross-encoder | 87.1 ± 0.1 | 84.8 ± 0.3 | 59.4 ± 0.4 | 67.4 ± 0.7 | 90.5 ± 0.3 | 75.6 ± 0.4 | 83.3 ± 0.4 | 82.8 ± 0.3 | 98.4 ± 0.1 | 89.4 ± 0.2 |
| Our pre-training on Toronto Books + Wikipedia | | | | | | | | | | |
| Bi-encoder | 84.6 ± 0.1 | 82.0 ± 0.1 | 54.9 ± 0.5 | 64.5 ± 0.5 | 88.1 ± 0.2 | 72.6 ± 0.4 | 80.9 ± 0.5 | 80.8 ± 0.5 | 98.4 ± 0.1 | 88.2 ± 0.4 |
| Poly-encoder (First-m) 16 | 84.1 ± 0.2 | 81.4 ± 0.2 | 53.9 ± 2.7 | 63.3 ± 2.9 | 87.2 ± 1.5 | 71.6 ± 2.4 | 80.8 ± 0.5 | 80.6 ± 0.4 | 98.4 ± 0.1 | 88.1 ± 0.3 |
| Poly-encoder (Learnt-m) 16 | 85.4 ± 0.2 | 82.7 ± 0.1 | 56.0 ± 0.4 | 65.3 ± 0.9 | 88.2 ± 0.7 | 73.2 ± 0.7 | 84.0 ± 0.1 | 83.4 ± 0.2 | 98.7 ± 0.0 | 89.9 ± 0.1 |
| Poly-encoder (First-m) 64 | 86.1 ± 0.4 | 83.9 ± 0.3 | 55.6 ± 0.9 | 64.3 ± 1.5 | 87.8 ± 0.4 | 72.5 ± 1.0 | 80.9 ± 0.6 | 80.7 ± 0.6 | 98.4 ± 0.0 | 88.2 ± 0.4 |
| Poly-encoder (Learnt-m) 64 | 85.6 ± 0.1 | 83.3 ± 0.1 | 56.2 ± 0.4 | 65.8 ± 0.7 | 88.4 ± 0.3 | 73.5 ± 0.5 | 84.0 ± 0.1 | 83.4 ± 0.1 | 98.7 ± 0.0 | 89.9 ± 0.0 |
| Poly-encoder (First-m) 360 | 86.6 ± 0.3 | 84.4 ± 0.2 | 57.5 ± 0.4 | 66.5 ± 1.2 | 89.0 ± 0.5 | 74.4 ± 0.7 | 81.3 ± 0.6 | 81.1 ± 0.4 | 98.4 ± 0.2 | 88.4 ± 0.3 |
| Poly-encoder (Learnt-m) 360 | 86.1 ± 0.1 | 83.8 ± 0.1 | 56.5 ± 0.8 | 65.8 ± 0.7 | 88.5 ± 0.6 | 73.6 ± 0.6 | 84.2 ± 0.2 | 83.7 ± 0.0 | 98.7 ± 0.1 | 90.1 ± 0.0 |
| Cross-encoder | 87.3 ± 0.5 | 84.9 ± 0.3 | 57.7 ± 0.5 | 65.3 ± 1.0 | 89.7 ± 0.5 | 73.8 ± 0.6 | 83.2 ± 0.8 | 83.1 ± 0.7 | 98.7 ± 0.1 | 89.7 ± 0.5 |
| Our pre-training on Reddit | | | | | | | | | | |
| Bi-encoder | 86.9 ± 0.1 | 84.8 ± 0.1 | 60.1 ± 0.4 | 70.9 ± 0.5 | 90.6 ± 0.3 | 78.1 ± 0.3 | 83.7 ± 0.7 | 83.6 ± 0.7 | 98.8 ± 0.1 | 90.1 ± 0.4 |
| Poly-encoder (First-m) 16 | 89.0 ± 0.1 | 86.4 ± 0.3 | 60.4 ± 0.3 | 70.7 ± 0.7 | 91.0 ± 0.4 | 78.0 ± 0.5 | 84.3 ± 0.3 | 84.3 ± 0.2 | 98.9 ± 0.0 | 90.5 ± 0.1 |
| Poly-encoder (Learnt-m) 16 | 88.6 ± 0.3 | 86.3 ± 0.3 | 61.1 ± 0.4 | 71.6 ± 0.6 | 91.3 ± 0.3 | 78.4 ± 0.4 | 86.1 ± 0.1 | 86.0 ± 0.1 | 99.0 ± 0.1 | 91.5 ± 0.1 |
| Poly-encoder (First-m) 64 | 89.5 ± 0.1 | 87.3 ± 0.2 | 61.0 ± 0.4 | 70.9 ± 0.6 | 91.5 ± 0.5 | 78.0 ± 0.3 | 84.0 ± 0.4 | 83.9 ± 0.4 | 98.8 ± 0.0 | 90.3 ± 0.3 |
| Poly-encoder (Learnt-m) 64 | 89.0 ± 0.1 | 86.5 ± 0.2 | 60.9 ± 0.6 | 71.2 ± 0.8 | 91.3 ± 0.4 | 78.2 ± 0.7 | 86.2 ± 0.1 | 85.9 ± 0.1 | 99.1 ± 0.0 | 91.5 ± 0.1 |
| Poly-encoder (First-m) 360 | 90.0 ± 0.1 | 87.3 ± 0.1 | 61.1 ± 1.9 | 70.9 ± 2.1 | 91.5 ± 0.9 | 77.9 ± 1.6 | 84.8 ± 0.5 | 84.6 ± 0.5 | 98.9 ± 0.1 | 90.7 ± 0.3 |
| Poly-encoder (Learnt-m) 360 | 89.2 ± 0.1 | 86.8 ± 0.1 | 61.2 ± 0.2 | 71.4 ± 1.0 | 91.1 ± 0.3 | 78.3 ± 0.7 | 86.3 ± 0.1 | 85.9 ± 0.1 | 99.1 ± 0.0 | 91.5 ± 0.0 |
| Cross-encoder | 90.3 ± 0.2 | 87.9 ± 0.2 | 63.9 ± 0.3 | 71.7 ± 0.3 | 92.4 ± 0.5 | 79.0 ± 0.2 | 86.7 ± 0.1 | 86.5 ± 0.1 | 99.1 ± 0.0 | 91.9 ± 0.0 |

Table 10: Validation and test performances of Bi-, Poly- and Cross-encoders. Scores are shown for ConvAI2, DSTC7 Track 1 and Ubuntu v2, and the previous state-of-the-art models in the literature.

