# OpenReview forum: "Poly-encoders: Architectures and Pre-training Strategies for Fast and Accurate Multi-sentence Scoring"
_ICLR.cc/2020/Conference — Accept (Poster)_

### Official Review · AnonReviewer2 · 2019-10-26
**Official Blind Review #2**

**Rating:** 8

**Review:**

This paper presents a new neural network architecture based on transformers called poly-encoders. These are compared against many state-of-the-art approaches including cross-encoders and bi-encoders on many large-scale datasets.
Bi-encoders > Poly-encoders > Cross-encoders in terms of speed and
Cross-encoders > Poly-encoders > Bi-encoders in terms of accuracy.

I am not an expert in this area. However, to the best of my knowledge I don't see anything immediately wrong with this. The experiments are also comprehensive. Therefore I recommend acceptance.

**Experience Assessment:**

I do not know much about this area.

**Review Assessment: Checking Correctness Of Derivations And Theory:**

N/A

**Review Assessment: Checking Correctness Of Experiments:**

I assessed the sensibility of the experiments.

**Review Assessment: Thoroughness In Paper Reading:**

I read the paper at least twice and used my best judgement in assessing the paper.

---

### Official Review · AnonReviewer1 · 2019-11-02
**Official Blind Review #1**

**Rating:** 6

**Review:**

This paper describes an approach for scoring sentences based on pre-trained transformers. The paper describes two main existing approaches for this task, namely bi-encoders and cross-encoders and then proposes a new formulation called poly-encoders which aims to sit between the existing approaches offering high accuracy -- similarly to cross-encoders -- and high efficiency -- similarly to bi-encoders. The paper is well written and although this is not related to my research I enjoying reading it. The approach proposed seems reasonable to me, and of sufficient novelty while the results presented are impressive. Moreover the paper seems a good fit for ICLR.

**Experience Assessment:**

I do not know much about this area.

**Review Assessment: Checking Correctness Of Derivations And Theory:**

N/A

**Review Assessment: Checking Correctness Of Experiments:**

I assessed the sensibility of the experiments.

**Review Assessment: Thoroughness In Paper Reading:**

I read the paper at least twice and used my best judgement in assessing the paper.

---

### Official Review · AnonReviewer4 · 2019-11-03
**Official Blind Review #4**

**Rating:** 8

**Review:**

Summary: This work proposes a new transformer architecture for tasks that involve a query sequence and multiple candidate sequences. The proposed architecture, called poly-encoder, strikes a balance between a dual encoder which independently encodes the query and candidate and combines representations at the top, and a more expressive architecture which does full joint attention over the concatenated query and candidate sequences. Experiments on utterance retrieval tasks for dialog and an information retrieval task show that poly-encoders strike a good trade-off between the inference speed of the dual encoder model and the performance of the full attention model.

Pros:
- Strong results compared to baselines on multiple dialog and retrieval tasks.
- Detailed discussion of hyperparameter choices and good ablations.
- Paper is well written and easy to follow.

Cons:
- Limited novelty of methods. Ideas similar to the model variants discussed in this work have been considered in other work (Eg: [1]). It is also known that in-domain pre-training (i.e, pre-training on data close to the downstream task’s data distribution) helps (Eg: [2]). So this work can be considered as an application of existing ideas to dialog tasks.
- In terms of impact, utterance retrieval has fairly limited applicability in dialog. The dialog tasks considered in this work have a maximum of 100 candidate utterances, whereas in practice, the space of possible responses is much larger. While retrieval models are useful, I am skeptical about the practical value of the improvements shown in the paper (especially the improvements over bi-encoder, which is already a decent model).

Suggestions:
One way to get around the inefficiency of the cross-encoder architecture is to first use an inexpensive scoring mechanism such as TFIDF or bi-encoder to identify a small number of promising candidates from all the possible candidates. We can then use the cross-encoder to do more precise scoring of only the promising candidates. I am curious where a pipelined model such as this compares against the variants discussed in the paper in terms of speed and performance.

While the paper presents strong results on several dialog utterance retrieval tasks, the methods presented have limited novelty and impact. I am hence leaning towards borderline.

References

[1] Logeswaran Lajanugen, Ming-Wei Chang, Kenton Lee, Kristina Toutanova, Jacob Devlin, and Honglak Lee. 2019. Zero-Shot Entity Linking by Reading Entity Descriptions. In Proceedings of the 57th Annual Meeting of the Association for Computational Linguistics.
[2] Jeremy Howard and Sebastian Ruder. 2018. Universal language model fine-tuning for text classification. In Proceedings of the 56th Annual Meeting of the Association for Computational Linguistics.

Edit: I have read the author response. Based on the rebuttal, I am more convinced about the practical impact of the approach. I am raising my score and recommending accept.

**Experience Assessment:**

I have published one or two papers in this area.

**Review Assessment: Checking Correctness Of Derivations And Theory:**

I carefully checked the derivations and theory.

**Review Assessment: Checking Correctness Of Experiments:**

I carefully checked the experiments.

**Review Assessment: Thoroughness In Paper Reading:**

I read the paper thoroughly.

---

> ### Author Response · Authors · 2019-11-14
> **Response**
>
> Thank you for your review.
>
> Re: "utterance retrieval has fairly limited applicability in dialog"
>
> Firstly, there are whole competitions run by dialogue researchers evaluating retrieval systems, e.g. DSTC7 Track 1 last year, and again this year in DSTC8, implying researchers in the field feel it is very important. Secondly, on a number of dialogue tasks, direct human evaluation comparing SOTA generative models with SOTA retrieval models ends up with retrieval models winning, see e.g. https://openreview.net/forum?id=r1l73iRqKm from last ICLR.
>
>
> Re: "The dialog tasks considered in this work have a maximum of 100 candidate utterances, whereas in practice, the space of possible responses is much larger."
>
> The tasks do really involve 100,000+ candidates (all utterances from the training set, see Table 1) but the evaluation metrics used in previous work involve using a subset of those per evaluated example, presumably because methods such as cross-encoder are too slow to evaluate otherwise. Poly-encoder can handle these sizes as evidenced in Table 5, which is of course the actual goal. Note the IR task evaluation did use 10,000 candidates also (Table 4), where the methods worked very well. This point seems to be more a criticism of standard evaluation practice, which we follow,  than our method.
>
>
> Re: "I am skeptical about the practical value of the improvements shown in the paper (especially the improvements over bi-encoder, which is already a decent model)."
>
> The anonymity constraint of ICLR makes this response harder than it should be for us to reply to -- in fact, this approach has become our standard method going forward that we use in real situations, and hence we do emphasize it has strong practical value. The method is practical because it is elegant & simple, fast and gives great results, as evidenced by the evaluation metrics (Table 4) and inference speed (Table 5). For us, it’s one of those papers where you do actually end up using the method, which definitely isn’t every time!
>
>
> Re: "One way to get around the inefficiency of the cross-encoder architecture is to first use an inexpensive scoring mechanism such as TFIDF or bi-encoder to identify a small number of promising candidates from all the possible candidates. We can then use the cross-encoder to do more precise scoring of only the promising candidates. I am curious where a pipelined model such as this compares against the variants discussed in the paper in terms of speed and performance."
>
> Building hybrids, pipelines and ensembles is often useful, but in this case looks tricky. For example, if we cut down the number of candidates from 100k to 1k with a bi-encoder, then switched to a cross-encoder, we would still need 20 seconds (on CPU) to rank with the cross-encoder (see Table 5). In this paper, we only compare single, related architectures against each other (bi, cross, poly).
>
>
> Re: "Ideas similar to the model variants discussed in this work have been considered in other work (Eg: [1])."
>
> Firstly, our work actually predates that work (an earlier version of this submission was uploaded to a non-archival venue). In any case, their brief description of architectures is not completely clear to us in terms of overlap, but apparently what they tried did not work as they conclude “The significant gap between Full-Transformer and the other variants shows the importance of allowing fine-grained comparisons between the two inputs via the cross attention mechanism embedded in the Transformer”. This is quite a different conclusion to ours, where we developed Poly-encoders which have almost the same performance as cross-encoders, but with huge speed-ups.
>
>
> Re: "It is also known that in-domain pre-training (i.e, pre-training on data close to the downstream task’s data distribution) helps (Eg: [2]). So this work can be considered as an application of existing ideas to dialog tasks."
>
> We agree that it is long-known that multi-tasking similar tasks is more useful than dissimilar tasks, as cited in our paper, and indeed our work is another example of this. As we understand [2], which is also on a different topic as you say, doesn’t actually compare two types of pre-training to show this helps though, it only compares “using no pre-training with pre-training on WikiText-103” (and then fine-tunes on the data of interest).  We also note that WikiText 103 experiments are not on the same scale as ours -- our pre-training on Reddit is more than 100x larger and compares to modern BERT pre-training.  We believe our result is important because much recent work is ignoring related-domain pre-training and using BERT-based (and variant) models etc. and scaling to larger & larger data without considering this crucial point of related-domain pre-training.  Our work provides clear empirical results that this is important even at massive scale.  (We will however add this cite, thanks.)

---

### Decision · Program_Chairs · 2019-12-19

**Decision:**

Accept (Poster)

**Comment:**

The paper presents a new architecture that achieves the advantages of both Bi-encoder and Cross-encoder architectures. The proposed idea is reasonable and well-motivated, and the paper is clearly written. The experimental results on retrieval and dialog tasks are strong, achieving high accuracy while the computational efficiency is orders of magnitude smaller than Cross-encoder. All reviewers recommend acceptance of the paper and this AC concurs.